# Parallel Analysis of Offshore Wind Turbine Structures under Ultimate Loads

**Shen-Haw Ju \*** , **Yu-Cheng Huang and Hsin-Hsiang Hsu**

Department of Civil Engineering, National Cheng Kung University, Tainan City 70101, Taiwan;
yuichen@mail.tainan.gov.tw (Y.-C.H.); overcomer_andy@hotmail.com (H.-H.H.)
**\*** Correspondence: juju@mail.ncku.edu.tw

**Abstract:** This paper investigates efficient design of offshore wind turbine (OWT) support structures under ultimate loads and proposes three schemes to overcome excessive computer time due to many required external loads. The first is the assumption of a rigid support structure to find blade wind forces, so that these forces are only dependent on wind profiles, which limits different cases in the structural analyses. Since the blade information is often confidential in turbine companies, this two-stage analysis allows the hub force to be the input data for the support structure design. The second is using a few control loads to perform the steel design between the second and the second-last design cycles. The third is using parallel computational procedures, since all loading cases can be independently executed in different CPU cores and computers. The test cases, with 5044 loading cases, indicate that the proposed method is fully parallel and can complete the design procedures using a few personal computers within several days. Test cases include IEC 61400-3, tropical cyclone, and seismic loads; although there are many loads to be considered, steel design is governed by a limited number of load cases, which are discussed in this paper.

**Keywords:** tropic cyclone; dynamic time-history analysis; offshore wind turbine; optimal steel design; parallel computation; support structure; ultimate load

## 1. Introduction

Optimization design and parallel computing technologies are important for the design of offshore wind turbine (OWT) support structures, because the design codes, such as IEC61400-3 [1], DNV-RP-C205 [2], and API [3], require a time–history dynamic analysis under many loading cases, for which the combinations of the structural analysis results are not suitable, due to the nonlinearity of soil. Moreover, the loading combinations of wave and wind loads require multi-directions, so the required loading cases can be greater than a thousand, which results in a significant amount of computer time. We will focus on the literature review for the optimal analysis or design of OWT support structures. Negm and Maalawi [4] described five optimization models for the design of a wind turbine structure, and their solutions showed significant improvements in overall system performance. The work of Anders et al. [5] presented a mechanical model of a wind turbine with momentum- and energy-conserving time integration, and numerical studies proved that physically consistent time-stepping schemes provide reliable results. The work of Christiansen et al. [6] investigated the influence of wind speed, wave frequencies, and misalignment between wind and waves for OWTs, and demonstrated a reduction in the structural oscillations, while improving power performance. The work of Borg et al. [7] studied the structural behavior of floating vertical axis wind turbines, and emphasized computational efficiency during the preliminary design stages. The work of Choi et al. [8] optimally designed a wind turbine system, with a minimum material cost, while the platform and substructure were optimized using a genetic algorithm. Muskulus [9] used a small set of loads, that together represented all possible worst-case scenarios, to simulate wind

turbines, and the results showed that this approach can be useful for jacket structures. The work of Lee et al. [10] established an optimal design process for the sub-structure of floating-type OWTs, using the neuro-response surface method, and confirmed the usefulness of the constructed framework in hydrodynamic performance. Yang and Zhu [11] presented a design optimization framework for OWT support structures, taking uncertainties into consideration. A numerical case of a tripod-type OWT was built for formal optimization. The work of Schafhirt et al. [12] performed structural optimization, assuming that changing the dimensions of a structural member does not affect other members, which allows for a simple sizing algorithm. The work of Gentils et al. [13] developed a structural optimization model for OWT, based on a coupled parametric finite element analysis and genetic algorithm, and the results showed a 19.8% reduction in the global mass of the support structure. The work of Yin et al. [14] studied the effects of blade pitch and rotor yaw, as well as wind-wave misalignment. The results showed that the effects of blade pitch and rotor yaw on turbine structural dynamics are significant, whereas the effect of wind–wave misalignment is small. The work of AlHamaydeh et al. [15] used a genetic algorithm to conduct structural design optimization for an OWT, and the results showed the proposed method to be superior in terms of finding optimal solutions. The work of Zuo et al. [16] used multiple tuned mass dampers to control vibrations from the fundamental and higher modes of an OWT tower, under combined wind, sea wave, and earthquake excitations. The work of Young et al. [17] presented an optimal design for a floating wind turbine tower, while the steel tower was compared to the composite materials tower, and the results demonstrated that the composite can significantly reduce the tower mass. Abhinav and Saha [18] investigated the response of a jacket supporting an OWT with the Define SSI, where the effect of the SSI became predominant for an OWT in loose sands. The work of Ju et al. [19] carried out ultimate load analyses, and determined the optimum design for a jacket-type OWT support structure under tropical cyclone and IEC loads. The work of Kim et al. [20] presented a nonlinear finite element model to simulate the interface behavior of a concrete grouted connection in a monopile OWT structure, and the results showed good agreement with the experimental results. The work of Plodpradit et al. [21,22] proposed the coupled dynamic analysis of the tripod and jacket structures for OWTs with piles–oil–structure interaction (PSSI), by using X-SEA and FAST v8 programs. The results showed that the support structure, considering PSSI, exhibited decreased natural frequencies and had a more flexible response, compared to the fixed-support structure. The work of Kim et al. [23] proposed a 3D OWT model on the prestressed concrete (PSC), supported by a pile foundation, which is suitable for weak soils. There are a number of computer programs available to solve the dynamics of OWTs within the time-domain, such as FAST [24], HAWC2 [25] and 3Dfloat [26] in academic arenas, and Bladed [27], Fedem Windpower [28], and ASHES [29] in commercial fields.

The optimal analysis and design of OWT support structures requires analyzing a significant number of loading cases, which results in immense computer time. This study is an attempt to solve this problem, using a two-step analysis method, where dominant loads between the first and last design cycles are selected to design OWT structures, and a parallel computational scheme using personal computers is also included. For the fatigue analysis and design of OWT support structures, a parallel computational method has been established [30]. Thus, this study mainly performs the parallel OWT structural analysis and design under ultimate loads. Both fatigue and ultimate load design Fortran programs can be found in the web site myweb.ncku.edu.tw/~{}juju, while the input data explanation can be found in the source codes.

## 2. Efficient Schemes for the Design of Offshore Wind Turbine Structures

In the structural design of bridges or buildings, the number of loading combinations is not more than 300 for most codes, such as AASHTO [31] and ASCE 7-98 [32], since the wind and seismic loads are arranged using their ultimate conditions. However, in the design codes of OWT structures, such as IEC61400-3 [1] and DNV-RP-C205 [2], the loading combinations of wave and wind loads require multiple directions, and must be able to act in the same or different directions, for both ultimate and service conditions. Thus, we proposed three schemes to increase the efficiency of the numerical computation, as follows:

(1)    The assumption of blade wind forces independent of the support structure

We assume that the time-dependent wind forces acting on the blades of a wind turbine are independent of the support structure, so these forces can be calculated at the hub center as three forces and three moments, considering only the blades, without the support structure. This scheme may be useful for real-world design procedures, since the blade information is often confidential in turbine companies, and the calculated hub forces can be used in two-stage analysis. This assumption may cause some deviation in analysis, but can significantly reduce computer simulation time. Since the six force components are the same when changing only the wind direction or wave force data (magnitude and frequency), this calculation can be omitted for many loading cases. For example, we generated a wind turbine structure analysis with 5044 loading cases, including seismic loads according to IEC 61400-3 when there are only 242 different time–history blade wind loads. It was found that this assumption can reduce computation time for blade wind loads, but the accuracy of this assumption is still in question. We used the NREL 5 MW wind turbine [33], with the jacket type foundation, to compare the difference between the two assumptions using Turbsim [34] and FAST [24], where the input data was obtained directly from reference [33] for the flexible support structure analysis, and the degrees of freedom of the tower and support structure were eliminated for the rigid support structure analysis. We analyzed the two conditions under two loading cases: a Design Load Case (DLC) 1.1 normal turbulence model with a $V_{hub}$ of 25 m/s, and a DLC 6.1 extreme wind speed turbulence model with a $V_{hub}$ of 47.5 m/s. It should be noted that DLC 1.1 is under the power production condition and DLC 6.1 is under the parked condition. Figure 1 shows the comparisons, which indicate that the results of the two assumptions were not very different. In reality, it is not correct to use the blade forces calculated from a flexible model to analyze the support structure, since the blade forces near the natural frequencies of the support structure are magnified, which causes resonance of the support structure and blade forces. One can omit this scheme to perform the full model analysis, but a considerably increased computer time will be required.

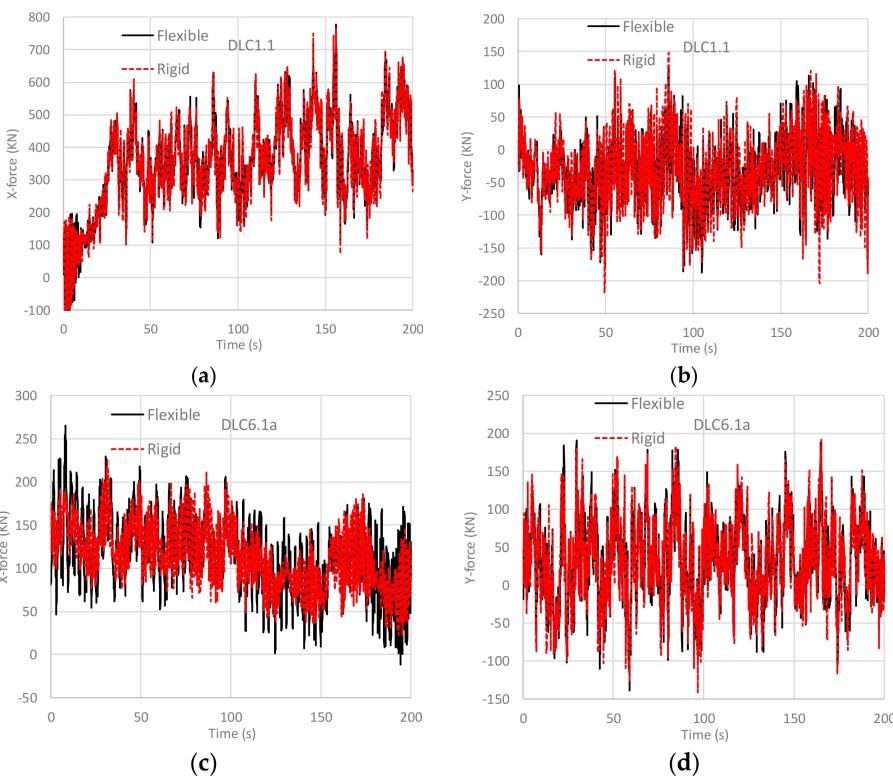

**Figure 1.** Comparisons of the total blade center forces between the rigid and flexible assumptions of the support structure under DLCs 1.1 and 6.1: (**a**) X-direction force of DLC 1.1 (**b**) Y-direction force of DLC 1.2 (**c**) X-direction force of DLC 6.1 (**d**) Y-direction force of DLC 6.1.

(2)    Using control loads to perform the design procedure between the second and second last design cycles

In the design procedure, one should first estimate the member sections and perform a structural analysis and steel design to find the appropriate section of each member. Then, the above procedure is performed continuously until all the selected sections are suitable and qualified under all the loads. This means that the structural analyses under all the loads should be analyzed several times. To save computer time, an alternative scheme is to select a few control loads between the second and the second-last design procedures (not including the first and final design cycles) to perform the structural analyses and steel designs. In this paper, we select the top $N_{load}$ maximum control loads of each member from the first design cycle, where $N_{load}$ is an input number, and combine the total intersection control loads of each member, to obtain all the necessary analyzed loads ($N_{total}$) between the second and the second-last design cycles. For example, if $N_{load}$ is set to 10, our proposed computer program will select the top ten maximum control loads of each member after the first design cycle, and use the intersection loads ($N_{total}$ loads) from all the members to perform the analysis and design between design cycle 2 to the second-last. In the final design cycle, all loads identical to design cycle 1 will be included. We use an OWT support structural design, with 1326 loading cases, without seismic loads, to test the efficiency and accuracy of this scheme, where the loads include IEC 61400-3 design situations (1) to (7), and there are 4710 degrees of freedom for the OWT support structure. The parameter $N_{load}$ is set at 10, 20, and 40, for which all the loads (1326), $N_{total}$ and computer time are shown in Table 1, which indicates that the required number of analyzed loads between the first and the final design cycle for $N_{load} \leq 40$ is much smaller than the total number of loads (1326). Moreover, the final design results for the four different $N_{load}$ are exactly the same. This table also indicates that, when one selects an appropriate $N_{load}$, the computer time can be less than half of that without this scheme. We suggest a $N_{load}$ ranging from 10 to 20 as an appropriate parameter. Using Table 1, we can estimate the computer time if the interaction analysis of the blades and support structure is used. When the $N_{load}$ is set at 10, using an Intel I7-6700K computer, the computer time is approximately 30.9 days, which is much longer than the 131.8 h, using the two-step design procedure, as discussed in (1).

(3)    Using parallel computational procedures

Because the design of OWT structures requires the time–history dynamic analysis of many loading cases, generated from such things as dead, wind, wave, and seismic loads, excessive computer time, for example, the computer time shown in Table 1, will cause a significant problem in the design work. Therefore, using parallel processing is the best method to overcome this problem, since all the loading cases can be independently executed, not only in different CPU cores, but also in different computers. However, it is necessary to establish a procedure to automatically arrange all the analysis and design results located in different CPU cores and computers, or too much manpower will be required.

**Table 1.** The necessary number of analyzed loads ($N_{total}$), and computer time between the second and the second last design cycles due to the change in the $N_{load,}$ using an Intel I7-6700K computer (there are five analysis and design cycles, while Turbsim and Fast programs are only required to execute the first cycle).

| Parameters and Computer Time | Case 1 | Case 2 | Case 3 | Case 4 |
|---|---|---|---|---|
| $N_{load}$ (Number) | 10 | 20 | 40 | 1326 |
| $N_{total}$ (Number) | 117 | 166 | 254 | 1326 |
| Turbsim and Fast, 117 cases (hour) | 24.7 | 24.7 | 24.7 | 24.7 |
| Structural analysis and design (hour) | 107.2 | 112.6 | 122.1 | 237.0 |

## 3. Optimal Parallel Ultimate Design Procedures for OWT Support Structures

This section discusses the optimal parallel ultimate design procedures for OWT support structures. In some regions, it is necessary to consider the ultimate loads from earthquakes or tropical cyclones. For seismic loads, we proposed a finite element framework for OWT support structures subjected to seismic, wind, and wave loads, including the soil–structure interaction and a conservative soil liquefaction analysis [35]. For tropical cyclones, this study includes two procedures—one accords to the GL Tropical Cyclone Technical Note (DLCs 11.2 and 11.3) [36] or IEC 61400-3 (DLCs I.1 and I.2) [1], and the other is based on the theory of a tropical cyclone passing the OWT support structure—to obtain the time–history turbulent wind distribution. Figure 2 shows the proposed optimal parallel design procedures for OWT structures under ultimate loads. The procedures are as follows: (1) Only an input file is required to define the structural dimensions, member sizes, loads, and the parallel computer arrangement. (2) A mesh generation program, Windturb, is used to generate a structural mesh with each loading case linked to a directory in a specific computer. (3) The TS, Turbsim, and Fast programs are then performed to find the distribution of the wind load, and the time–history blade forces and moments, where TS, developed in this paper, calculates the wind loads using IEC or DNV-GL codes, and moreover, the non-stationary turbulent wind of tropical cyclones can be generated [19]. Since we assume a rigid support structure to find the blade forces, only a small proportion of the loading cases have to run in the two programs. Moreover, each analysis is executed in a CPU core independently, so this step is fully parallel. (4) After the analyses in Step 3 are finished, the structural analysis and steel design of each loading case are then performed in each directory, using a CPU core, and this step is also fully parallel. (5) An optimal program is executed to find the appropriate thickness of each member from all the structural analysis results, and the total weight of the structure is calculated. (6) If all the member sizes are acceptable, the design procedure is terminated, or the new member sizes are used and, Step 2 is initiated to perform the next design cycle. However, the Turbsim and Fast programs for Step 3 do not have to be run again, since the blade forces are the same.

In the above procedures, steps (1), (2), (5), and (6) cannot be parallel, but they only use a very small amount of computer time, since the major computer time is spent on Steps (3) and (4), which are in the fully parallel computation. Figure 3 shows the proposed finite element model and applied loads of the OWT support structures. The 3D two-node beam element is used to model the columns, bracings, piles, and tower, and the four-node plate element is used to model the platform at the bottom of the tower. The p–y, t–z, and Q–z curve springs, according to the API formulations [3], are used to model the soil–structure interaction. The static loads, such as dead and live working loads, are applied and analyzed using a one-step static analysis at the beginning. The dynamic loads, such as the wind, wave, and seismic loads, are analyzed using Newmark's direct integration method with the Raleigh damping scheme. The Turbsim program is used to find the 3D wind distribution, and the Fast program is used to find the time–history of blade forces and moments at the hub center. The wave loads are both regular and irregular; the stream function theory is used to find the regular wave load, and the second-order function theory [37] is used to determine the irregular wave load. The seismic load uses the time and depth dependent displacements applied to one end of each p–y, t–z, and Q–c curve spring, where the Shake91 program [38] is used to find the depth dependent acceleration when the seismic accelerations at a specific depth are known. Since the p–y, t–z, and Q–c curve springs are nonlinear, the Newton–Raphson method is used to solve this nonlinear matrix equation. Using the above procedures, as shown in Figure 2, the designer can give the approximate member sizes of the support structure, and the optimal program will obtain the suitable thickness of each member after several design cycles. Figure 4 shows the total steel mass of the 5-MW OWT support structure using three groups of initial member section data, for which the design results are very similar after five design cycles.

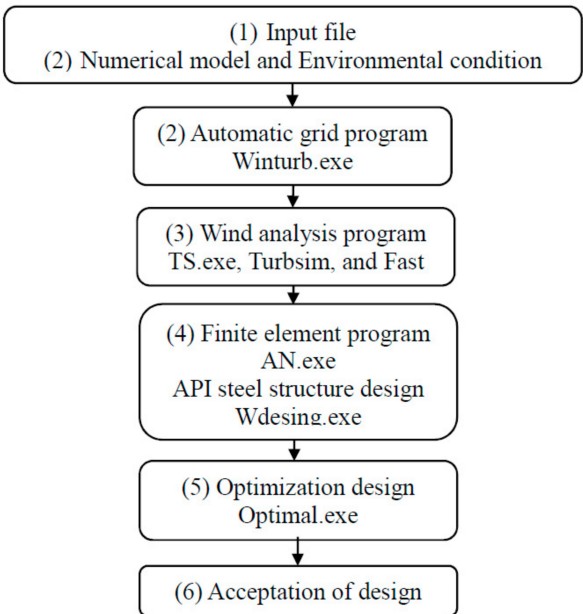

**Figure 2.** Optimal design flowchart for offshore wind turbine (OWT) support structures.

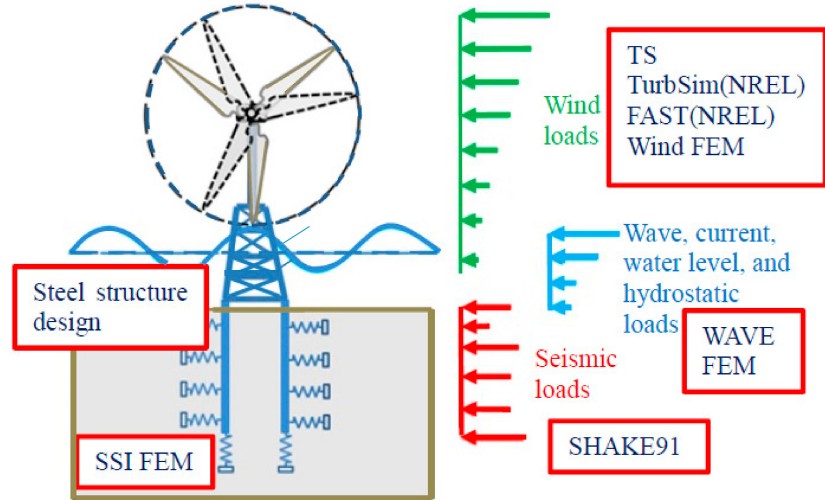

**Figure 3.** Finite element model and loads for the optimal design procedure for OWT support structures.

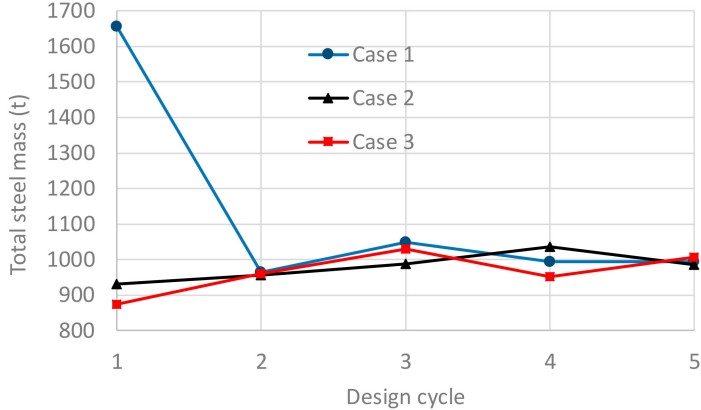

**Figure 4.** The total mass of the 5-MW OWT support structure at each design cycle, using three groups of initial member section data.

Due to the large number of DLCs, it is necessary to extend the parallel processing technique in a single machine to multi machines to speed up the efficiency of analyses. We proposed a simple method to perform the parallel computation in Steps (2) and (3), shown in Figure 2 using multi computers. The procedures are listed as follows:

(1) In computer one (the main computer), the user first defines the disk name, number of CPU cores, and CPU efficiency of each computer in an input file. In addition, all the input data, such as structure and loading information, can be also arranged in this file for convenience, so that there is only one input file in the main computer.

(2) The Windturb mesh generation program arranges the number of Turbsim and Fast analyses and the number of structural analyses and designs in each computer, according to the efficiency of each computer. Then, the program arranges the input file with the loading cases required to run in each computer.

(3) In each computer, a batch program is executed in the same manner as using a single computer to perform the procedures shown in Figure 2.

(4) At the end of each design cycle (Step (6) in Figure 2), the batch program in each computer finds the minimum required thickness of each member in the current computer and is then paused. When all the computers are at the waiting step, the batch program in the main computer finds the minimum required thickness of each member from all the computers and changes the appropriate member thicknesses in each computer.

(5) The batch program of each computer begins to perform the analysis and design for the next cycle, until the delivery of satisfactory design results.

The above procedures can be fully parallel if the efficiency of each computer is set appropriately. For example, one can use several computers that are the same model, so the computer efficiency is the same for each computer. Only Step (4) requires communication between computers. However, in this step, only a small file from each computer is read, using very little computer time. The source codes, manual, and examples, including fatigue and ultimate designs, can be accessed at myweb.ncku.edu.tw/~{}juju/.

## 4. Study of 10 MW OWT under IEC 61400-3 Loads with Earthquake and Typhoon

The DTU 10-MW OWT [39], with the jacked type support structure, is analyzed as an example in this section, where the average depth of the sea is 50 m, and the soil is moderate-hard sandy soil with a submerge internal frictional angle of 30°. IEC 61400-3 (2019) loads, with the exception of the fatigue and transport loads (DLC 8), are analyzed, where earthquake and typhoon effects are included. Three earthquakes, with a peak ground acceleration (PGA) of 0.32 g and $T_s = 1$ s, were also generated, and eight earthquake directions were set at 0° to 315°, with an increment of 45°, from the X axis, where $T_s$ is the period at the end of constant acceleration in the response spectrum [40]. We assumed that the soil liquefaction occurs 20 m from the sea-bed during the earthquake, and the analysis followed the reference [35]. Moreover, the IEC 61400-3 DLC I.1 and DLC I.2 are used to simulate a class-three tropical cyclone, where the average wind speed at the hub height is 72 m/s, and the significant wave height of a return period of 500 years is 14 m. The finite element mesh and the dimensions of the final design result are shown in Figure 5, where the mesh contains 176 beam elements for columns, 256 beam elements for bracings, 28 beam elements for the tower, 280 beam elements for piles, 36 plate elements for the platform, and 280 p–y curve elements, 280 t–z curve elements, and four Q–z curve elements for soil simulation. In the analysis and design, there are 5044 applied loads listed in Table 2 and the total degree of freedom of the OWT support structure is 4710. For the cases without seismic and tropical cyclone loads, the time step length was set at 0.05 s, and 14,000 time steps (total analysis time = 700 s) were simulated using the direct integration Newmark's method. For the cases with seismic loads, the time step length of 0.01 s, and 18,000 time steps, were used (total analysis time = 180 s), where the seismic load began at 100 s and lasted for 40 s. Rayleigh damping, with a mass damping of 0.04/s and stiffness damping of 0.01 s was used, which produces a damping ratio near 4%

at frequencies of 0.3 and 4 Hz. Table 3 shows the computer time, executed in finite element design programs, using one to eight computers. Tables 4–7 show the required steel weight of the design and the control loads of the tower, columns, braces, and piles for various conditions, including IEC 61400-3, class-3 typhoon, or 0.32-g earthquake loads. These results indicate the following features:

(1)  The assumption of rigid support structures, to obtain the blade forces and moments to perform the structure analysis, can reduce computer time by nearly three times the amount of the full analysis model, which considers the interaction of blades, support structures, and wind. Moreover, the design of OWT support structures can be fully parallel using independent computers, and the computer time required for the communication between computers is negligible, and can be ignored.

(2)  As shown in Tables 4–7, the number of necessary loading cases can be over one thousand, but most of these are not dominant in the design of member sizes. Thus, the design cycles between the second and the second-last cases can use a number of control loads, and we suggest finding control loading cases ranging from ten to twenty critical loads for each member ($10 \leq N_{load} \leq 20$) is appropriate in the design of OWT support structures.

(3)  Without seismic and typhoon loads, DLC 2.2 may dominate the steel design for the tower and columns. The reason for this is that the blades are out of control at a small instantaneous wind speed, at which the pitch angle is small to obtain wind energy, and the larger turbulent wind later exerts a large force on the blades. Except for this condition, the parked condition, with a large wind speed and wave (DLCs 6.1 and 7.1) will dominate the OWT steel design.

(4)  An earthquake with a PGA of 0.32 g and Ts of 1 s is similar to the designed seismic load of buildings and bridges in Taiwan, where a strong seismic zone is located, and the 20 m soil liquefaction is also a conservative assumption. Tables 4 and 5 indicate that the required steel weight increases about 5%, compared to the IEC 61400-3 loads, due to this. The major controlled parts are pile and brace members, while the column and tower members are still dominated by the IEC 61400-3 loads.

(5)  Class-3 tropical cyclones may occur near the Pacific and Atlantic regions. Table 6 indicates that, due to this, required steel weight increases about 19%, compared to the IEC 61400-3 loads. Almost all steel design is controlled by IEC 61400-3 DLC I.2, in which the yaw misalignment is an important factor; this condition is similar to the results discussed by Ju et al. [19], according to GL Tropical Cyclone Technical Note [36].

(6)  If one needs to include the class-3 tropical cyclone load and the seismic load, with the PGA of 0.32 g and 20 m soil liquefaction, the required steel weight increases about 21% compared to the IEC 61400-3 loads, as shown in Table 6. The structure is still dominated by the tropical cyclone load of DLC I.2, but some of the bracings and piles are controlled by seismic loads. The reason for this is that the 20 m soil liquefaction causes the piles and bracings to weaken.

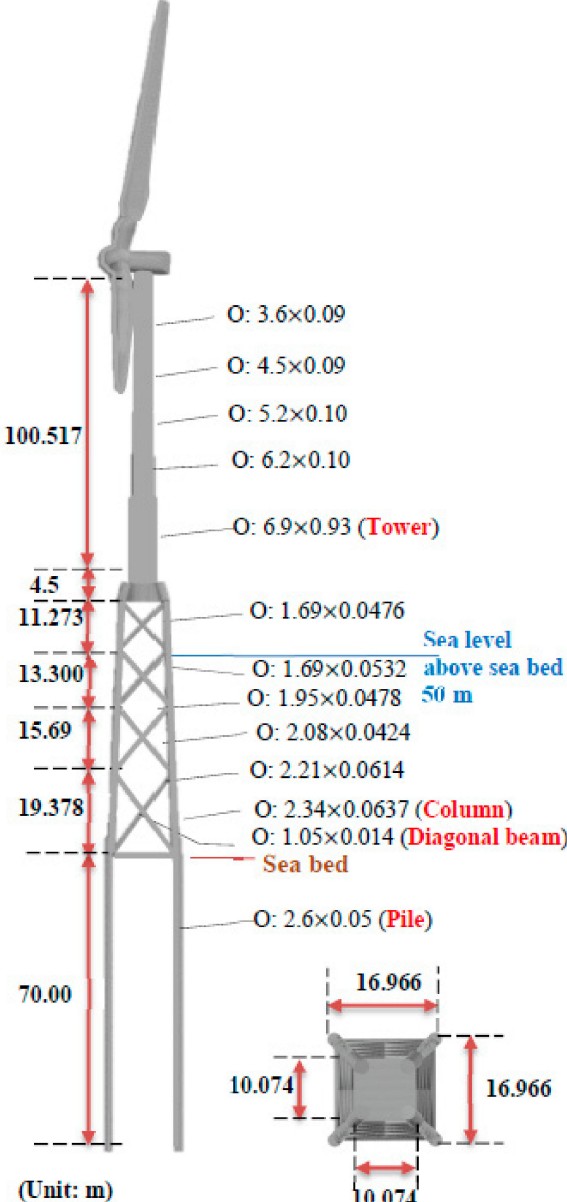

**Figure 5.** The finite element mesh, dimensions, and final section sizes for the design, under the 0.32-g seismic load and class-3 tropical cyclone load ($V_{hub}$ = 72 m/s).

**Table 2.** Illustration of the 5044 applied loads in the structural design (the abbreviated words are referred to in IEC 61400-3 (2019)).

| Design Situation | DLC | Wind Condition $V_{hub}$ (m/s) | | Waves (m) | | Wind Dir. / Wave Dir. | Yaw | Partial Safety Factor |
|---|---|---|---|---|---|---|---|---|
| Power production | 1.1 | NTM | 3 to 25 interval 2 | NSS | 0.47 to 3.88 interval 0.31 | 0° / 0° | 0° | 1.25 |
| | 1.3 | ETM | DLC 1.1 | NSS | DLC 1.1 | DLC 1.1 | 0° | 1.35 |
| | 1.4 | ECD | 9.4, 11.4, 13.4 | NSS | 1.6, 1.8, 2.1 | ECD / 0° | 0° | 1.35 |
| | 1.5 | EWS | DLC 1.1 | NSS | DLC 1.1 | DLC 1.1 | 0° | 1.35 |
| | 1.6 | NTM | DLC 1.1 | SSS | DLC 1.1 | DLC 1.1 | 0° | 1.35 |

**Table 2.** *Cont.*

| Design Situation | DLC | Wind Condition $V_{hub}$ (m/s) | | Waves (m) | | Wind Dir. Wave Dir. | Yaw | Partial Safety Factor |
|---|---|---|---|---|---|---|---|---|
| Power production plus occurrence of fault | 2.1 | NTM | DLC 1.1 | NSS | DLC 1.1 | DLC 1.1 | 0° | 1.35 |
| | 2.2 | NTM | DLC 1.1 | NSS | DLC 1.1 | DLC 1.1 | 0° | 1.1 |
| | 2.3 | EOG | 9.4 to 13.4 interval 0.5 and 25 | NSS | 1.6 to 2.1 interval 0.0625 & 3.88 | DLC 1.1 | 0° | 1.1 |
| | 2.5 | NWP | DLC 1.1 | NSS | DLC 1.1 | DLC 1.1 | 0° | 1.35 |
| Start-up | 3.2 | EOG | 3, 9.4 to 13.4 interval 0.5 and 25 | NSS | 0.47, 1.6 to 2.1 interval 0.0625 & 3.88 | DLC 1.1 | 0° | 1.35 |
| | 3.3 | EDC | DLC 3.2 | NSS | DLC 3.2 | ECD 0° | 0° | 1.35 |
| Normal shut down | 4.2 | NWP | DLC 2.3 | NSS | DLC 2.3 | DLC 1.1 | 0° | 1.35 |
| Emergency shut down | 5.1 | NTM | DLC 2.3 | NSS | DLC 2.3 | DLC 1.1 | 0° | 1.35 |
| Parked (standing still or idling) | 6.1 | EWM (T) | 57 | ESS ($H_{s50}$) | 12.7 | Note$_{Ang}$ Note$_{Ang1}$ | 0°, 4°, 8° | 1.35 |
| | 6.2 | EWM (T) | DLC 6.1 | ESS ($H_{s50}$) | DLC 6.1 | Note$_{Ang}$ Note$_{Ang2}$ | 0° to 180° interval 10° | 1.1 |
| | 6.3 | EWM (T) | 45.6 | ESS ($H_{s1}$) | 9 | DLC 6.2 | 0°, 10°, 20° | 1.35 |
| Parked and fault conditions | 7.1 | EWM (T) | DLC 6.3 | ESS ($H_{s1}$) | 9 | DLC 6.2 | DLC 6.2 | 1.1 |
| | 7.1A | EWM (T) | DLC6.3 | ESS ($H_{s1}$) | 9 | DLC 6.2 | 0°, 4°, 8° | 1.1 |
| Parked (standing still or idling) | I.1 | EWM (T) | $V_{10min.500} = 72$ | ESS | 14 | DLC 6.2 | 0° | 1.0 |
| | I.2 | EWM (T) | $V_{10min.500} = 72$ | ESS | 14 | DLC 6.2 | 0° to 180° interval 30° | 1.0 |
| Power production (Earthquake plus grid loss) | 1.8 | NTM | $V_{hub} = V_r = 13.4$ | NSS | 1.6 | DLC 1.1 | 0° | 1.0 |
| Parked (Earthquake plus grid loss) | 6.7 | NTM | $V_{hub} = 0.7$ $V_{ref} = 39.9$ | NSS | $H_s = 0.7$ $H_{s50} = 8.89$ | DLC 1.1 | 0° | 1.0 |

Note: (1) (T) = Turbulent wind model. (2) $V_{hub}$ = average wind speed at hub height. (3) $H_s$ = significant wave height. (4) Note$_{Aag}$ = 0° to 180° with an interval of 30°. Note$_{Aag1}$ = −30°, −15°, 0°, 15°, and 30° relative to the wind direction. (5) Note$_{Aag2}$ = −30°, 0°, and 30° relative to the wind direction. (6) DLC 2.1 = uncontrolled pitch angle at 300 s. (7) DLC 2.2 = uncontrolled pitch angle at 310, 315, and 320 s for three conditions. (8) DLC 7.1A = fixed pitch angle at 0°, 15°, 30°, 45°, and 60° for five conditions. (9) Parked condition = fixed pitch angle at 90°. (10) DLC 1.8 = three different earthquakes from the directions of 0°, 45°, 90°, 135°, 180°, 225°, 270°, and 315° under the controlled blade pitches and uncontrolled blade pitches, after 15 s of the seismic load. (11) DLC 6.7 = the same as DLC 1.8 with the always parked blade pitches. (12) The wave period is set to 11.1 $\sqrt{H_s/g}$, 12.2 $\sqrt{H_s/g}$, 13.2 $\sqrt{H_s/g}$, and 14.3 $\sqrt{H_s/g}$.

**Table 3.** Computer time for the structural analysis and design changes, with the number of Intel I7-6700K computers used in a parallel computation (there are five analysis and design cycles with $N_{load}$ = 10 and $N_{total}$ = 213 in each computer, while Turbsim and Fast programs are only required to execute the first cycle).

| Number of Computers | 1 | 2 | 3 | 4 | 6 |
|---|---|---|---|---|---|
| Structural analysis and design (hour) | 204 | 102 | 69 | 52 | 35 |

**Table 4.** The control loads of the analyzed structure at the last design cycle for only IEC 61400-3 loads (total required steel weight of the design = 2168 t).

| Element | Control Cases | Wind Speed (m/s) | $H_s$ or $H$ (m) | Direction Wind/Wave/Yaw | Weight Ratio (%) |
|---------|---------------|------------------|------------------|-------------------------|------------------|
| Column | DLC 2.2 | 15 | 2.3 | 0°/0°/0° | 56 |
| | DLC 6.1 | 57 | 12.7 | 0°/−30°/8° | 44 |
| Brace | DLC 6.1 | 57 | 12.7 | 30°/0°/8° | 39 |
| | DLC 6.1 | 57 | 12.7 | 0°/−15°/8° | 28 |
| | DLC 6.1 | 57 | 12.7 | 30°/15°/8° | 17 |
| | DLC 6.1 | 57 | 12.7 | 0°/−30°/8° | 15 |
| | DLC 6.1 | 57 | 12.7 | 0°/0°/8° | 1 |
| Tower | DLC 2.2 | 15 | 2.3 | 0°/0°/0° | 94 |
| | DLC 7.1 | 45.6 | 9 | 0°/−30°/8° | 5 |
| | DLC 7.1 | 45.6 | 9 | 60°/30°/0° | 2 |
| Pile | DLC 6.1 | 57 | 12.7 | 0°/−30°/8° | 53 |
| | DLC 6.1 | 57 | 12.7 | 180°/210°/0° | 47 |

**Table 5.** The control loads of the analyzed structure at the last design cycle for IEC 61400-3 and earthquake loads (total required steel weight of the design = 2278 t).

| Element | Control Cases | Wind Speed (m/s) | $H_s$ or $H$ (m) | Direction Wind/Wave/Yaw | Weight Ratio (%) |
|---------|---------------|------------------|------------------|-------------------------|------------------|
| Column | DLC 2.2 | 15 | 2.3 | 0°/0°/0° | 56 |
| | DLC 6.1 | 57 | 12.7 | 0°/−30°/8° | 44 |
| Brace | DLC 6.7 | 39.9 | 8.9 | 0°/0°/0° | 52 |
| | DLC 6.1 | 57 | 12.7 | 0°/0°/8° | 34 |
| | DLC 6.1 | 57 | 12.7 | 0°/−30°/8° | 12 |
| | DLC 6.1 | 57 | 12.7 | 0°/−15°/8° | 1 |
| | DLC 6.1 | 57 | 12.7 | 30°/0°/8° | 1 |
| Tower | DLC 2.2 | 15 | 2.3 | 0°/0°/0° | 94 |
| | DLC 7.1 | 45.6 | 9 | 30°/0°/4° | 6 |
| Pile | DLC 6.7 | 39.9 | 8.9 | 0°/0°/0° | 39 |
| | DLC 6.7 | 39.9 | 8.9 | 0°/0°/0° | 27 |
| | DLC 6.1 | 57 | 12.7 | 0°/−30°/8° | 24 |
| | DLC 1.8 | 11.4 | 1.8 | 0°/0°/0° | 11 |

**Table 6.** The control loads of the analyzed structure at the last design cycle for IEC 61400-3 and typhoon loads (total required steel weight of the design = 2582 t).

| Element | Control Cases | Wind Speed (m/s) | $H_s$ or $H$ (m) | Direction Wind/Wave/Yaw | Weight Ratio (%) |
|---------|---------------|------------------|------------------|-------------------------|------------------|
| Column | DLC I.2 | 72 | 14 | 120°/30°/30° | 76 |
| | DLC I.2 | 72 | 14 | 90°/30°/60° | 24 |
| Brace | DLC I.2 | 72 | 14 | 0°/0°/30° | 97 |
| | DLC I.2 | 72 | 14 | 30°/0°/30° | 3 |
| Tower | DLC I.2 | 72 | 14 | 90°/60°/60° | 52 |
| | DLC I.2 | 72 | 14 | 30°/0°/60° | 28 |
| | DLC I.2 | 72 | 14 | 90°/30°/60° | 14 |
| | DLC I.2 | 72 | 14 | 30°/30°/90° | 5 |
| | DLC 7.1 | 45.6 | 9 | 30°/0°/0° | 2 |
| Pile | DLC I.2 | 72 | 14 | 120°/30°/30° | 76 |
| | DLC I.2 | 72 | 14 | 0°/180°/30° | 24 |

**Table 7.** The control loads of the analyzed structure at the last design cycle for IEC 61400-3, typhoon, and earthquake loads (total required steel weight of the design = 2622 t).

| Element | Control Cases | Wind Speed (m/s) | $H_s$ or $H$ (m) | Direction Wind/Wave/Yaw | Weight Ratio (%) |
|---------|---------------|------------------|------------------|-------------------------|------------------|
| Column | DLC I.2 | 72 | 14 | 120°/30°/30° | 61 |
| | DLC I.2 | 72 | 14 | 90°/30°/60° | 24 |
| | DLC I.2 | 72 | 14 | 120°/60°/30° | 14 |
| Bracing | DLC I.2 | 72 | 14 | 0°/0°/30° | 50 |
| | DLC 6.7 | 39.9 | 8.9 | 0°/0°/0° | 48 |
| | DLC I.2 | 72 | 14 | 30°/0°/30° | 2 |
| Tower | DLC I.2 | 72 | 14 | 90°/30°/60° | 39 |
| | DLC I.2 | 72 | 14 | 90°/60°/60° | 36 |
| | DLC I.2 | 72 | 14 | 30°/0°/60° | 19 |
| | DLC I.2 | 72 | 14 | 30°/30°/90° | 5 |
| | DLC 7.1 | 45.6 | 9 | 60°/0°/0° | 2 |
| Pile | DLC I.2 | 72 | 14 | 120°/30°/30° | 58 |
| | DLC 1.8 | 11.4 | 1.8 | 0°/0°/0° | 24 |
| | DLC 6.7 | 39.9 | 8.9 | 0°/0°/0° | 9 |
| | DLC I.2 | 72 | 14 | 120°/60°/30° | 8 |

## 5. Conclusions

Over one thousand analyzed loading cases can be required to design OWT support structures using combinations of various loads, which uses a significant amount of computer time. In this paper, three schemes were proposed to overcome this difficulty. The first is the assumption of blade wind forces independent of the support structure, so forces can be calculated at the hub center as three forces and three moments, considering the blades without the support structure. This assumption can reduce computer simulation time significantly, since the six components are only dependent on wind profiles, which form only a limited number of different cases in the structural analyses. Moreover, the components are the same for each design cycle. The second scheme is using a few control loads to perform the design procedure between the second and second last design cycles. From the test cases, 50% of total computer time can be saved, compared to without this scheme, but the final design results are exact the same. The third scheme involves the use of parallel computational procedures, since all the loading cases can be independently executed in different CPU cores, as well as on different computers. We established an efficient procedure, so that manpower was minimal. The test cases, using 5044 loading cases, indicated that the proposed method can complete the design procedures using a number of personal computers within several days, and the design procedures are fully parallel.

The DTU 10-MW OWT support structure was analyzed with 5044 loads, including IEC 61400-3, tropical cyclone, and seismic loads; however, the steel design was controlled by several loads. Without seismic and typhoon loads, DLC 2.2 may dominate the steel design for the tower and columns, and DLCs 6.1 and 7.1, the parked condition with large wind speed and wave, may dominate steel design for bracings and piles. For earthquakes which are not extreme, such as the PGA of 0.32 g with 20 m soil liquefaction, the required steel weight increases about 5%, compared to the IEC 61400-three loads. The major controlled parts are piles and bracings, while the tower and columns are still dominated by the IEC 61400-3 loads. For class-3 tropical cyclones, the required steel weight increases by about 19%, compared to the IEC 61400-3 loads, and almost all the steel design is controlled by IEC 61400-3 DLC I.2, where yaw misalignment is important factor. If the class-3 tropical cyclone load and the seismic load with the PGA of 0.32 g and 20 m soil liquefaction are considered, the required steel weight increases about 21%, compared to the IEC 61400-3 loads. The structure is still dominated by the tropical cyclone load of DLC I.2, but some of the bracings and piles are controlled by seismic loads. The reason for this is that the 20 m soil liquefaction causes the piles and bracings to weaken.

**Author Contributions:** S.-H.J. was responsible for simulations and paper writing, and Y.-C.H. and H.-H.H. checked the measured data and helped analyze results.

**Funding:** This research received no external funding.

**Conflicts of Interest:** The authors declare no conflict of interest.

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
