# Peer review of "Parallel Analysis of Offshore Wind Turbine Structures under Ultimate Loads"

_applsci, doi:10.3390/app9214708_

Round 1

Reviewer 1 Report

This is a useful paper because there is a considerable amount of work required to design a wind turbine support structure and this paper looks at several ways of reducing that workload.

I attach a file marked up with suggested revisions.

Author Response

Please see the PDF file.

Reviewer 2 Report

Dear Authors, 

Thank you for submitted your manuscript. Even so I believe that the topic you presented is of interest, I think that the manuscript needs a major revision.

In general, the manuscript lacks details of the models (e.g. DOF etc.) and by far a more depth presentation of the results and analysis/discussion of the data. The second part of the manuscript, where you applied the method to the DTU 10MW OWT, is significantly better described but also here details and results are missing. 

Author Response

Please see the PDF file.

Round 2

Reviewer 2 Report

Dear Authors, 

Congratulations, the manuscript has significantly improved. I attached the PDF with minor comments. Same of the figure have a poor quality/low resolution. Please increase the quality of the figure! 

Author Response

The Reviewer’s time and effort are greatly appreciated. We feel the comments have resulted in an enhanced manuscript. Our responses are as below:

The word “If” changes to “if”.

ANS: We appreciate the correction from the reviewer, and have corrected this error.

Congratulations, the manuscript has significantly improved. I attached the PDF with minor comments. Same of the figure have a poor quality/low resolution. Please increase the quality of the figure! 

ANS: We appreciate the comment from the reviewer, and have modified the image resolutions of figures 3 and 5.